# Spatial Barriers to Transforming toward a Healthy Food System in the Noreste of Mexico

**DOI:** 10.3390/nu16091259

**Published:** 2024-04-24

**Authors:** Rob Roggema, Aleksandra Krstikj, Brianda Flores

**Affiliations:** 1School of Architecture, Art and Design, Campus Monterrey, Tecnologico de Monterrey, Monterrey 64849, Mexico; 2School of Architecture, Art and Design, Campus State of Mexico, Tecnologico de Monterrey, López Mateos 52926, Mexico; sandra.krstik@tec.mx; 3School of Humanities and Education, Campus Monterrey, Tecnologico de Monterrey, Monterrey 64849, Mexico; a00829567@tec.mx

**Keywords:** food environments, plant-based diets, traditional gastronomy, food security, urban agriculture

## Abstract

In the past five decades, global food systems have undergone a notable transition, moving from predominantly rural settings to increasingly urban and industrialized environments, largely driven by processes of globalization and supply chain integration. However, this evolution has not adequately addressed equitable access to nutritious diets and food environments, resulting in adverse health outcomes. This study delves into the spatial and non-spatial barriers that impede the adoption of healthy diets in the Noreste of Mexico, particularly focusing on the challenges associated with accessing and cultivating plant-based foods. Through an examination of suitable areas for urban agriculture and an exploration of the socio-cultural factors influencing the adoption of plant-based diets, the research focuses on interventions aimed at promoting healthier and more sustainable eating practices in Monterrey. The findings of the study reveal significant disparities in food access across the Monterrey metropolitan area, with central urban zones exhibiting superior access to fresh foods compared to suburban and peripheral regions. This inequality disproportionately affects marginalized areas characterized by higher poverty rates, exacerbating issues of food insecurity. Nevertheless, traditional dietary practices could offer promising avenues for creating culturally significant and healthier dietary transitions, even amidst the ongoing process of urbanization.

## 1. Introduction

Over the last five decades, food systems have evolved from rural to industrialized and urban settings, driven by increased globalization and supply chain integration [1]. While this evolution has improved access to food for an ever-expanding and diverse global population, it has also sparked significant concerns about dietary patterns with low-quality nutrition [2]. In conjunction with this trend, the consumption of ultra-processed foods containing excessive amounts of salt, sugar, and trans fats [3] is increasing, while traditional food practices are gradually disappearing [4]. Meanwhile, fewer people are preparing meals at home with fresh ingredients. Notably, nutritional deficiencies and imbalanced diets worldwide have resulted in severe health concerns, such as high rates of obesity, cardiovascular problems, diabetes, and other non-communicable diseases (NCDs), collectively responsible for 74% of all global deaths [5].

Food systems currently fail to provide a balanced and nutritional diet equitable to all, which negatively impacts physical and mental health [6]. One-third of the worldwide population suffers from malnutrition in some form, a highly prevalent public health concern in countries with low or middle incomes. If existing patterns persist or worsen, this number might increase to half by 2025 [7]. Symptoms of anxiety disorders, depression, and other mental health issues are also directly related to nutrient deficiencies in diets and inadequate access to food [8]. Moreover, the immediate outcomes of existing food systems, influencing food availability, quality, safety, and affordability, are incompatible with global sustainability and detrimental to several environmental processes [9]. Industrial agriculture is responsible for increasing carbon dioxide emissions, biodiversity loss, natural resource degradation, and ecosystem destruction [10].

UN agencies and other international organizations promote healthy diets to prevent food insecurity and chronic diseases [11]. This dietary transition requires changes to consumer behavior, food environments, and supply chains to ensure adequate nutrition and environmental sustainability within planetary boundaries. Nevertheless, despite sustainable recommendations in dietary guidelines across many countries, persistent ambiguity prevents their successful implementation and widespread acceptance [12]. In response to this challenge, the Eat-Lancet Commission introduced in 2019 the Planetary Health Diet, a pioneering framework informed by robust scientific evidence. According to these guidelines, this approach targets more sustainable dietary behaviors by prompting the consumption of plant-based foods, encompassing whole grains, legumes, nuts, fruits, and vegetables, complemented by moderate dairy consumption and minimal or no meat intake [13]. A typical EAT-Lancet diet has 2500 calories per day.

Empirical evidence supports the potential for dietary changes on a large scale to promote human health and environmental sustainability [14,15]. Incorporating healthy dietary patterns characterized by a diversified intake of vegetables, fruits, and legumes, alongside restricted or absent consumption of meat and dairy products, has been identified as a promising approach to mitigating greenhouse gas emissions (GHGE) [16]. Such dietary regimens, including an important intake of fruit and vegetables, have shown the potential to diminish premature mortality associated with NCDs and age-related disabilities [17] through adequate nutrient intake, which promotes weight loss and weight maintenance [18]. Furthermore, researchers suggest a healthier diet transition should involve reducing the intake of nutrient-deficient and energy-dense foods [19] by supporting food biodiversity and the associated traditional knowledge and practices that rely on it. [20] Recent decades have witnessed a burgeoning global interest in traditional diets, driven by the recognition of their longstanding presence and potential health benefits [21].

Even though guidelines and empirical evidence provide shared directions, context-specific approaches are crucial for fostering sustainable diets [17]. A sustainable diet is defined by the Food and Agriculture Organization (FAO) as one that “is protective and respectful of biodiversity and ecosystems, culturally acceptable, accessible, economically fair and affordable; nutritionally adequate, safe and healthy; while optimizing natural and human resources” [21]. Research has shown that locally based approaches, considering geographical features, climate circumstances, cultural customs, industrial methods, and regulatory contexts, foster healthier and more sustainable dietary behaviors [22,23]. Integrating these aspects into dietary guidelines can create more effective strategies that address different populations’ specific socio-cultural needs and preferences [24]. Notably, food environments play a pivotal role in dietary behavior, promoting nutritious food options that are culturally acceptable, locally available, and easily accessible [25]. Therefore, when formulating population-level dietary changes to promote environmental sustainability, human health, and societal well-being, it is crucial to identify and address any cultural and geographical barriers that may hinder implementation and long-term adherence [26].

Mexico is now experiencing a significant nutritional shift influenced by Western diets, resulting in unhealthy eating habits [27,28]. Consequently, energy-dense food intake has increased, while traditional staples such as whole grains and legumes have declined [29]. Only a minimal proportion of Mexican adults regularly consumes healthy foods, whereas most of the national population follows less healthy Western diets [30]. Consequently, chronic conditions such as type 2 diabetes, obesity, hypertension, and different types of cancer are on the rise nationwide and have become a major public health issue [31]. This nutritional transition, exacerbated by globalization and trade agreements, underscores the urgent need for comprehensive interventions to address food security and environmental challenges [32]. While the transition toward sustainable food systems in Mexico has recently received academic attention [6,27,28,29,30,33], there is still a considerable gap in knowledge regarding the context-specific state of food environments and diets.

Recent studies show the central role of local agency and sustainability in advocating for healthy and sustainable food choices, especially for marginalized and vulnerable populations [34,35]. A strong agency empowers individuals and communities to defend their access to food, advocate for localized and environmentally responsible agri-food systems, demand transparency from producers and policymakers, and make informed decisions about food consumption. Additionally, it supports community-led initiatives like farmers’ markets and community gardens, promoting nutrition education and local food production. Facilitating individual and community participation in decision making could increase involvement and commitment to healthier sustainable practices. This collaborative approach facilitates local leadership and promotes culturally relevant solutions, ultimately driving meaningful and enduring change in food systems.

Nuevo León contributes only 1.3% to national agricultural production, despite enjoying diverse environmental conditions conducive to crop cultivation. However, the challenge lies in the sparse rural population density, with only 4.1% of the population living in those areas. With fewer people engaged in agricultural activities, the focus shifts to other economic sectors. Over the past decade, a notable shift has occurred within the metropolitan area of Nuevo León toward the development of food production. Community-driven and entrepreneurial urban farming initiatives have taken place to improve food security and local food production. Besides growing different vegetables and cooking herbs, they have also been implemented for environmental education and social cohesion. However, without clear policies or incentives from local authorities, urban farming ventures have faced complex challenges in securing the resources, land access, and regulatory support needed to prosper, resulting in a discontinuity in urban farming in Monterrey.

In this work, we propose a systemic and contextual approach to mitigating nutritional-related health risks in communities. Previous research has already defined a sustainable diet and has developed methods for the spatial analysis of food environments. However, little research integrates spatial and non-spatial analysis that considers a variety of geographical features, climate circumstances, cultural customs, industrial methods, and regulatory contexts to foster healthier and more sustainable dietary behaviors. In an attempt to bridge this gap in knowledge, we developed a new framework that integrates the analysis of spatial and socio-cultural barriers into plant-based diets in the specific context of the metropolitan area of Monterrey in Mexico, and proposed guidelines for minimizing those barriers in public policy and planning. The spatial barriers were identified in terms of difficulty of access and the local production of plant-based food. Thus, the types of food environments were analyzed based on the availability of plant-based foods in outlets and walkable access from housing to an outlet. Moreover, the barriers to the local production of plant-based foods were analyzed through a preliminary research of suitable open area, soil type, and annual rainfall. The analysis of spatial barriers gave us insights that enabled us to identify primary and secondary zones in the metropolitan area of Monterrey where urban and periurban agriculture can have the most local impact.

We also aimed to understand the non-spatial barriers hindering plant-based diet adoption in Monterrey. Our study examined three distinct dietary patterns: the traditional Mexican diet, the EAT-Lancet diet tailored to Mexico, and a traditional Noreste of Mexico diet. Our analysis focused on assessing the nutritional value and sustainability of these diets. Considering the significant socio-cultural factors influencing food choices, we recognize the inherent challenges in transitioning to a plant-based diet. Our findings may ultimately serve as a basis to develop culturally appropriate strategies to promote healthier and more sustainable eating habits in Monterrey.

## 2. Materials and Methods

To investigate socio-cultural barriers to sustainable and healthy eating habits in the Noreste of Mexico (Nuevo León), we conducted a comprehensive study employing a methodologically rigorous approach. This involved examining three diets—(a) the traditional Mexican diet, (b) the EAT-Lancet diet tailored for Mexico, and (c) a diet based on regional dishes—from both environmental and health perspectives. We outlined each diet’s key ingredients and nutritional characteristics and discussed their potential health benefits based on existing literature. Leveraging existing literature and conducting a thorough dietary component analysis, we identified and scrutinized obstacles to implementing these diets in the Noreste region, considering both nutritional adequacy and sustainability considerations.

For the proposed diet with regional dishes of the Noreste, we engaged directly with the local community, collecting recipes from traditional female cooks and chefs in the metropolitan area of Monterrey. Through semi-structured interviews, the participants provided invaluable insights into ingredient selection, portion sizes, cooking techniques, and the historical background and cultural context surrounding local dishes. We categorized the frequency of ingredients in the collected recipes and visually represented the findings to better understand the traditional dietary patterns of Nuevo León. This approach allowed us to identify staple items within each food group outlined in the EAT-Lancet guidelines. By recognizing the predominant dietary components used in regional cuisine, deeply rooted in the biocultural food heritage of Nuevo León, we aimed to provide context-sensitive information for policymakers to formulate strategies that not only align with current health standards but also resonate with the cultural identity of the population.

In the spatial part of the analysis, we first analyzed the clustering of fresh food stores in the metropolitan area of Monterrey and the marginalization index per neighborhood. We used the census data of population from 2020 and the census of economic units from 2023 available from the National Institute of Statistics and Geography (INEGI) of Mexico [36]. This preliminary analysis enabled us to understand which populations had higher access to fresh food. To determine the type of food environment in each municipality, we used indicators of availability and accessibility to food in general and to fresh food in particular. We note that among outlets that sold fresh food we included shops that sold meat and dairy products, thus the actual availability of plant-based products is in fact overestimated in this scenario. In this research we did not include informal markets since previous research found that informal markets have a limited role in bringing plant-based foods closer to the population in Monterrey [37]. Informal markets in Mexico are open or partially open temporary street markets with vending stalls organized in rows. They are a common feature in Mexican cities, appearing once or twice per week in most neighborhoods. In Central Mexico, informal markets contribute significantly to bringing fresh fruits and vegetables to citizens; however, in Monterrey they are flea markets with a limited offer of fresh food products [37].

We estimated the availability of fresh food in each municipality as a ratio between population and number of food stores, where up to 100 people/store was considered high availability, 100 to 200 people/store was considered medium availability, and more than 200 people/store was considered low availability. The accessibility in each municipality was taken from the open data base of the Observatorio de Ciudades de Tecnologico de Monterrrey and was expressed in average walking time to reach a store [38]. In this study, the measure of up to 15 min walking to a store was considered high accessibility, medium accessibility was considered between 20 and 30 min, and more than 30 min was considered low accessibility. Based on these estimates, we divided the municipalities into three groups as follows:Food deserts—municipalities with low availability of any type of food and low accessibility;Food swamps—municipalities with medium or low availability of fresh food, high availability of processed food, and medium or good accessibility;Food oasis—municipalities with high availability of fresh food and high accessibility.

After the types of food environments were estimated, we discussed the data in relation to the prevalence of diet-related diseases in the population, such as overweight and obesity and cardiovascular diseases. The data on the prevalence of these diseases was obtained for the period 2022–2023 directly from the Institute of Social Security (IMSS) clinics in Monterrey. The correlation was used as a descriptive indicator to estimate trends and the location of places with higher risk of developing diet-related diseases.

Furthermore, we reviewed the suitability of areas for agricultural activity. We first estimated the area of each municipality that was unbuilt and had less than a 20% slope, the type of soil that predominated, and the annual rainfall in two periods: dry season November–April and wet season May–September. The unbuilt area was calculated as a rough estimate; from the total municipal area we extracted all built areas consisting of communication and infrastructure, housing, and other functional land uses. Since there is no precise data about communication or other services and facilities, we estimated 30% of the total area for communication and infrastructure. For services and facilities, we estimated an area equal to the housing area, based on the mixed-use planning tool proposed by van den Hoek in 2008 [39]. Most municipalities have underdeveloped services, communication and infrastructure systems, and housing predominates, thus our estimate was the most conservative scenario leaving aside sufficient space for a desirable urban structure where mix-use and accessibility would be significantly improved. Based on data available on the open interactive platform for hydrological research SIATL from INEGI [40], we revised dominant soil type and annual rainfall in each municipality. Thus, we estimated which municipalities had better conditions in terms of cultivation area per citizen, soil type for cultivation, and water availability, and assigned an index of suitability for urban agriculture as follows:Very high, where the unbuilt land is equal or larger than the built area, the soil type is excellent or good for cultivation, and water supply is sufficient or manageable for cultivation of some plant foods;Medium, where the unbuilt land is equal or larger than the built area, and either there is good soil type for cultivation or sufficient water supply—the index 2A signifies good soil type but insufficient water supply, while index 2B signifies soil type not optimal for cultivation but sufficient or manageable water supply;Low, where both water and soil demands are not met for cultivation.

In this categorization, we did not include areas that had a low proportion of unbuilt areas, even if they had good soil quality, since those areas usually have populations with higher socioeconomic classes and the urban agriculture outputs would be low due to lack of arable lands. With the above categories we aimed to identify areas where urban agriculture projects could have more significant impacts on more vulnerable communities in the urban periphery.

Finally, based on the socioeconomic structure of the population, including density and poverty rates, we discussed in which municipalities the support for urban agriculture focused on plant production could have higher social and health impact. We aimed to propose a spatial framework for public policies in each municipal region to maximize the positive outcomes for communities while promoting local and community-based urban or periurban agricultural initiatives. Therefore, considering the identified spatial barriers of access and production of plant-based foods in the metropolitan area of Monterrey, we proposed a division into primary and secondary areas where urban and periurban agriculture could have the most local impact. Primary areas were identified as those that were food deserts, had a significant percentage of urban poverty, and had good natural conditions to support agricultural activities. The secondary areas were identified in food swamps that also had considerable urban poverty but either lacked a sufficient population or natural conditions to have high local impact. The rest of the zones were considered as places where urban/periurban agriculture was not likely to be impactful. The aim of this typology was to serve as a base for designing tailored strategies to support plant-based production and consumption for each municipality.

## 3. Results

### 3.1. Diets

#### 3.1.1. The Traditional Mexican Diet

The traditional Mexican diet boasts different staple foods that contribute to its nutritional diversity, shaping the culinary identity of Mexico. According to Valerino-Perea [41], several key food groups have been identified as part of the traditional Mexican diet. These include mostly grains and tubers, legumes, and vegetables, with specific items like maize, beans, chili, squash, tomatoes, and onions being prominent. Although other food groups like fruits, beverages, meats, sweets and sweeteners, spices, herbs, and condiments are listed, they appear less frequently. Within this dietary framework, maize-based products such as tortillas and tamales serve as a foundational food of Mexican cuisine. Tubers like potatoes, sweet potatoes, and beans contribute significantly to dietary fiber intake. A wide variety of fruits, including indigenous options such as capulin and tejocote, in addition to globally consumed options such as pineapple and citrus, provide essential vitamins and minerals. While dairy products like cheese and yogurt are not typically part of this diet, poultry such as turkey and chicken, along with game meats like ducks, venison, and beef, serve as primary protein sources. In certain regions, traditional dishes may incorporate insects like grasshoppers and locusts, which offer protein, fat, and vitamins. Additionally, herbs and condiments like chili, epazote, and onion enhance flavor profiles and offer digestive benefits. Healthy fats are derived from sources like avocados and peanuts, while chia seeds contribute fiber and omega-3 fatty acids. Sweeteners like honey, sugar, sugarcane, and agave are used in moderation. Overall, the traditional Mexican diet emphasizes plant-based ingredients with limited meat consumption and no dairy products (Table 1).

#### 3.1.2. Eat-Lancet in Mexico

Based on the findings presented by Castellanos-Gutiérrez et al. [29], the adaptation of a healthy reference diet (EAT-HRD) recommendation to the Mexican context involved adjustments to align with the mean daily energy intake of Mexican adults, which stands at 1947 kcal/day, contrasting with the global recommendation of 2500 kcal/day. The study revealed that Mexican adults consumed grains at a level 1.3 times higher than the EAT-HRD reference, yet approximately one-third of this intake consisted of low-fiber grains and grains with excessive sugar or saturated fat. Notably, high-fiber grains, akin to whole grains, were consumed at approximately 0.7 times the recommended intake. To enhance grain consumption, emphasis should be placed on high-fiber grains like whole corn products, while the intake of refined grains and those with excess additives should be limited. Regarding vegetables and fruits, Mexican adults fell short of the EAT-HRD recommendations, consuming these at approximately 0.8 and 0.7 times the recommended amounts, respectively, with notably low consumption of dark green vegetables. Animal-based protein sources were consumed excessively, particularly red meat, which was consumed at 5.4 times the reference, whereas fish intake fell below the recommended level. Additionally, legumes and nuts were consumed at lower levels than recommended. Dairy intake exceeded recommendations, while added sugars intake was notably higher, and consumption of added fats was below the recommended level. In comparison with the Mexican Dietary Guidelines (MDGs), Mexican intake deviated significantly from recommendations, particularly in the shortfall of fruits, vegetables, legumes, and added fats. EAT-HRD adaptation to Mexico emphasizes high-fiber grains, fruits, vegetables, and legumes while reducing refined grains, added sugars, and excessive animal-based proteins. Tailoring recommendations to the Mexican context requires addressing socioeconomic disparities and cultural dietary preferences to improve overall nutritional quality and promote better health outcomes.

A further study exploring adherence to the EAT-Lancet Healthy Reference Diet in Mexico offers new insights while confirming previous findings [42]. While the participants generally adhered to specific dietary components, likely influenced by economic or cultural factors, concerns about elevated saturated fat consumption persist, as observed in earlier research. Additionally, despite moderate poultry, eggs, dairy, and fish meat consumption, this exceeded EAT-Lancet intake recommendations for over 80% of the sample, indicating a prevalent dietary trend (Table 2). Notably, gender disparities emerged, with men exhibiting higher fruit scores but lower scores for vegetables, tubers, and red meat than women. Socio-cultural factors, potentially related to gender roles in diet, may contribute to these differences.

#### 3.1.3. The Traditional Recipes Diet

The traditional diet of Nuevo León, located in the Noreste of Mexico, is a blend of diverse culinary influences shaped by historical and cultural exchanges. Indigenous, Spanish, and Jewish culinary traditions contribute distinct flavors and techniques to the regional cuisine, resulting in a rich and varied gastronomic heritage [43]. Meat is often the foundational component of traditional dishes, reflecting the area’s historical reliance on livestock farming. The study analyzed 25 collected recipes from Nuevo León using the EAT-HDR food group framework to understand the nutritional content of the local diet of the Noreste de Mexico. The methodological approach involved systematically categorizing ingredients based on their respective food groups to discern patterns and trends in recipe composition.

Our analysis reveals that meat is consistently the primary ingredient in traditional dishes, except in recipes where cacti (cabuche and nopal) constitute the main component. In our systematic analysis, beef (various animal parts) is the primary protein source, followed by pork and eggs. Notably, chicken is not mentioned in traditional recipes, but recent studies highlight chicken as the predominant protein in contemporary consumption patterns [44]. This shift underscores the evolving gastronomic landscape, potentially influenced by internal migrations and broader cultural influences. However, further research is necessary to fully comprehend the socio-cultural factors driving the integration of traditional foods from other regions into Nuevo León’s contemporary diets, providing valuable insight into local food traditions’ evolution.

We also could identify the use of different types of chilis in the collected recipes, such as serrano, ancho, jalapeño, cascabel, bell pepper, guajillo, and piquín—an endemic wild variety from the region. These are integral to the creation of salsas, which incorporate onions, garlic, tomatoes, and various herbs (including oregano and parsley) and spices (such as paprika and pepper). The nutritional value of salsas has recently been highlighted in recent studies, shedding light on their potential health benefits within a balanced diet [45]. Salsa is often used as an ingredient in meat preparation, with fats such as pork lard, beef tallow, and oil commonly employed in their cooking process. It is worth noting that in the traditional Nuevo León diet, potatoes dominated over beans as a complement to meat dishes. This observation suggests that traditional Nuevo León recipes rely less on plant-based protein sources than other Mexican cuisines.

On the other hand, fruits like lemons, red and green tomatoes, nopal (prickly pear cactus), cabuche (biznaga flower), and fresadilla tomatoes are essential in Nuevo León cuisine for their acidic flavor, a defining characteristic of Nuevo León cuisine, as noted in interviews with local chefs. While dairy products such as asadero cheese, cream, and butter are present, these products do not figure prominently in the recipes. Instead, animal-based protein sources include eggs, pork, goat, and beef, prepared in various cuts using traditional cooking methods like grilling, braising, or stewing. The traditional recipes collected generally eschew the use of sugars. However, when sugars are needed, there is a notable preference for non-centrifugal sugar (NCS), an unrefined or minimally refined product derived from sugarcane juice evaporation, namely brown sugar and piloncillo (sugar cane), which contrasts with modern sweet consumption practices.

The traditional Noreste diet emphasizes animal-based ingredients, notably lean beef (carne de agostadero), which provides a reliable protein source. Accompanied by staple ingredients like tomatoes, onions, garlic, and chilis, it supplies essential vitamins and minerals such as vitamin C, potassium, and antioxidants. These nutrients support immune function, regulate blood pressure, and combat oxidative stress. Additionally, various herbs and condiments enhance flavor without calories. With an emphasis on low sugar intake, this diet stabilizes blood sugar levels and reduces the risk of chronic conditions like diabetes. The cacti preparations further enrich the diet with dietary fiber, vitamins, and minerals. The consumption of less meat and a greater intake of whole grains and vegetables is recommended by EAT-HDR. While lean meats offer vital protein, excessive consumption may harm cardiovascular health. The traditional Noreste diet (Figure 1) can also be enhanced nutritionally by incorporating more plant-based options and a balance of protein sources.

In summary, our analysis of non-spatial barriers to a healthy food system in Monterrey revealed that the traditional Mexican diet emphasizes plant-based ingredients with limited meat consumption and no dairy products. Mexicans consume a lower amount of the global recommendation for kcal/day. Even though Mexican adults consume grains at a level 1.3 times higher than the EAT-HRD reference, approximately one-third of this intake consisted of low-fiber grains and grains with excessive sugar or saturated fat. Mexican adults fell short of the EAT-HRD recommendations regarding vegetable and fruit consumption, while animal-based protein sources were consumed excessively, particularly red meat, which was consumed at 5.4 times the reference. Gender disparities emerged in the analysis, with men exhibiting higher fruit scores but lower scores for vegetables, tubers, and red meat than women. In Nuevo León, meat—especially lean beef—is consistently the primary ingredient in traditional dishes, except in recipes where cacti (cabuche and nopal) constitute the main component. Lemons, red and green tomatoes, cabuche (biznaga flower), and fresadilla tomatoes are essential, as well as onions, garlic, and chilis.

### 3.2. Spatial Barriers

We found that in the metropolitan area of Monterrey, outlets that sell fresh food are mostly clustered in the central urban areas such as the municipalities of Monterrey, San Nicolás de los Garza, General Escobedo, and San Pedro Garca García (Figure 2). The suburban municipalities of Apodaca and Guadalupe also seem to have smaller clusters of fresh food outlets; however, the presence of such outlets diminishes with distance from the urban center. The peripheral municipalities where fresh food outlets are lacking seem to be the ones with a higher degree of poverty and marginalization, such as García, El Carmen, General Zuazua, Santa Catarina, and Santiago. This indicates a fragile context in the metropolitan periphery of Monterrey that can fuel further food access insecurity.

In Table 3, we estimate the types of food environments based on the availability of food outlets and walkable access. We found that in Monterrey, five of the sixteen urban municipalities and one of the four periurban municipalities can be considered food deserts. The most alarming cases seem to be García, since it has almost 400,000 people living in a food desert, and General Zuazua, where reaching a food outlet by foot is extremely difficult. Six municipalities in the urban context and three in the periurban context can be considered food swamps. Food swamp conditions were also found in the municipality of Guadalupe, where the number of processed food outlets is greater than that of fresh food outlets. Only four municipalities seem to have conditions of a food oasis. However, these four municipalities also house a high number of processed food outlets, but the high accessibility is a significant alleviating condition. Only one municipality, Juárez, was placed between the predefined categories of food swamp and food oasis and was labeled as “swampish” since despite its low accessibility, it houses a relatively low number of processed food outlets compared to fresh food outlets.

In Figure 3, we map the food deserts in Monterrey, which are predominantly found in the peripheral municipalities in the north, while the food swamps are in the suburban and periurban municipalities of the south. The food oases are grouped in the central and most dense area of Monterrey. Concerning diet-related diseases and food environments, we found that the clinic that reported the highest incidence of diet-related diseases in the period 2022–2023 is in the municipality of Monterrey, close to the border with Guadalupe, which is a food swamp. We previously categorized the municipality of Monterrey as a food oasis since many fresh food outlets are located there and the accessibility is high. Nevertheless, we also noted that there is a significant number of processed food stores as well and these are surrounded by food swamps. Thus, the reason might be circumstantial or cultural—one of choice. The limited local production of food in that municipality or lack of recreational facilities might be contributing factors as well. Thus, the relationship between the food environment and diet-related diseases is not straightforward and needs more contextual research. However, the clinic that reported the highest rise in cases of diet-related diseases is found in the transition area between Monterrey and General Escobedo, which indicates that the hot-spot of diet-related diseases is moving to the north of the metropolitan area where food deserts prevail. This could be an additional risk for the northern municipalities that already have high poverty rates and drought risks.

In Table 4 and Figure 4, we present the preliminary estimation of municipalities that have suitable conditions for developing urban/periurban agriculture. We found that the best conditions are presented in the municipalities of Cadereyta Jiménez, Pesquera, and Santiago due to their high percentage of unbuilt areas, fertile soils, and higher than average annual rainfall. The conditions seem especially good in the peripheral municipality of Santiago. The municipalities El Carmen, García, Santa Catarina, Hidalgo, and Abasolo (2A) have fertile soils but relatively low unbuilt area and a lack of annual rainfall to predict good agricultural harvests. The municipalities General Zuazua, Juárez, and San Nicolás de los Garza (2B) receive more than average annual rainfall but the cultivation area is small and their soils are not conducive to agricultural production. Finally, the municipalities of Apodaca, Ciénega de Flores, Monterrey, General Escobedo, San Pedro Garca García, Guadalupe, and Salinas Victoria are unsuitable for local production of plant-based foods since their soils are not fertile and they receive very low annual rainfall. That is to say that urban agriculture in these municipalities is not impossible but requires strategies for dry farming or space-intensive vertical gardening that can use tools such as water reutilization and hydroponics. It is notable that one food desert, the municipality of Pesquería, is one of the most suitable areas for plant-based local food production. The other food deserts are also suitable with either good soil or sufficient area and rainfall, except for the municipality of General Zuazua. We estimate that the eradication of the food desert problem in General Zuazua will be difficult through urban agriculture methods. 

Considering the previous analysis of the suitability of urban/periurban agriculture and the socioeconomic indicators, we estimated areas where urban/periurban agriculture could have most health and social impact (Table 5 and Figure 5). We found that the promotion and support of the local production of plant-based foods in the municipalities of Pesquería and Caderayta Jiménez could have the highest impact, since the suitability for such activities in terms of open arable land and the percent of people who live in poverty in these food environments are high. The impact could be especially significant in Pesquería since it is currently a food desert. The planning should support the urban agriculture initiatives by improving accessibility in Pesquería. Supporting urban/periurban agriculture in the other food deserts could also have a significant impact, especially in El Carmen and García, which have populations with the highest poverty index but also very fertile soils. These two municipalities are close to the IMSS clinic that reported the highest rise in diet-related diseases, thus urban agriculture there could improve fairness and equity in food access and health. Nevertheless, the agricultural strategies in El Carmen and García should be oriented to dry farming and the harvesting of water to promote better outcomes. On the other hand, farming in San Nicolás de los Garzas, which uses soil-enriching techniques, could promote the improvement of urban health in the zone where high rates of diet-related diseases are found. Santiago and Santa Catarina—dense and impoverished food swamps—should prioritize the support of plant-based local food production since the conditions are conducive to urban farming in these zones. Planning in Santiago should focus on improving access so that urban agriculture can have more impact through better food distribution. Unfortunately, urban/periurban agriculture in the densest municipalities such as Apodaca and Monterrey is not possible due to infertile soils and a lack of unbuilt areas, thus the promotion of such efforts there could have low social impacts unless they are focused on vertical farming and hydroponics. In Figure 6, we present a list of plant-based foods that could be promoted for urban agriculture in each municipality based on minimum water requirements for their cultivation and natural conditions. for the high impact zones that could serve as a guideline for future planning to increase plant-based diets.

Our analysis of the spatial barriers to a healthy food system in the Monterrey metropolitan area sheds light on the disparities in food availability and food accessibility in this area. While the central urban zones show high availability and accessibility of fresh foods, the peripheral municipalities exhibit low values of these indices and six of those peripheral municipalities are food deserts. These results indicate the urgent need to analyze city-planning land uses and mobility in relation to food outlets in the periphery of Monterrey. The health clinics that present the highest rates of diet-related diseases are located at the limits of food oasis and food swamps, indicating local-scale areas that warrant more disaggregated data research to understand the variety of factors that lead to this health outcomes. The lack of health clinics in peripheral municipalities might be skewing these results, thus further research is necessary to shed light on the health conditions of vulnerable populations living in food deserts on the periphery. Our analysis of the suitability for urban agriculture indicated the south-western municipalities as potentially the most suitable. Nevertheless, most of the population that currently lives in food deserts is in the north-eastern part of the metropolitan zone. This finding indicates the need for public policy that can develop metropolitan-scale urban agriculture planning and a strategy with a focus on logistics and the last-mile delivery of fresh local produce to counter these disparities in access to locally produced plant-based food.

### 3.3. Non-Spatial Barriers

In Monterrey, transitioning to a plant-based diet poses challenges shaped by socioeconomic status, biocultural diversity, dietary trends, and urbanization. Once a national staple, the traditional Mexican diet, rich in maize, vegetables, and legumes, has lost popularity among the population in recent decades, particularly in urban areas, leading to the loss of knowledge and practices related to traditional Mexican cuisine. This departure from historical dietary practices complicates the establishment of plant-based diets as cultural preferences develop and evolve. As urban lifestyles become more prevalent, especially among the young, there is an increased reliance on Western food consumption patterns, marked by a high consumption of sugar, salt, saturated fats, and red meat [27].

Moreover, socioeconomic factors compound this complexity, as income inequalities and resource access shape dietary preferences and purchasing power, potentially limiting the adoption of plant-based diets among lower-income populations in Monterrey. For instance, individuals with higher socioeconomic status often consume animal-based foods (primarily meat) driven by social and economic factors [47]. Conversely, individuals with lower socioeconomic status typically incorporate fewer animal proteins, including dairy, eggs, fish, or meat, into their diets [30]. Disadvantaged population groups in Mexico may be more inclined to adhere to traditional dietary patterns centered around staples like maize and beans. However, in Monterrey the scarcity of affordable fresh food options in informal street markets (tianguis) [37] may contribute to the heightened consumption of highly processed and industrialized foods among lower socioeconomic households, exacerbating health risks associated with an excessive intake of sugary beverages, high-fat processed foods, and those laden with added sugars, thus posing a significant barrier to the adoption of healthier eating patterns.

Mexico’s food consumption patterns vary significantly across regions [42]. For instance, the Noreste region exhibits substantial meat consumption driven by its long-established livestock farming traditions, while crop production output remains relatively modest. Notably, Nuevo León’s contribution to national agricultural production is only one percent [48], hindering the adoption of sustainable plant-based diets as most vegetables and fruits must be imported. The city of Monterrey, located within this geographical area, confronts environmental challenges owing to its semi-arid climate in local agricultural production, which limits plant-based alternatives as an economically feasible alternative. As urbanization advances in Monterrey, compounded by limited agricultural productivity in Nuevo León, adherence to Western dietary practices becomes further entrenched, impeding the adoption of plant-based diets. In contrast, regions such as the central and southern areas, characterized by traditional farming systems like the milpa, are more likely to incorporate maize, fruits, and legumes into their daily diets, facilitating a healthier and more sustainable eating pattern. Effectively promoting plant-based dietary transitions in Mexico, particularly in regions characterized by minimal agricultural heritage and pronounced meat consumption, necessitates tailored interventions addressing these multifaceted barriers, which include cultural sensitivity and affordable plant-based alternatives.

In sum, the traditional Mexican diet, rich in maize, vegetables, and legumes, has lost popularity among the population in recent decades, leading to the loss of knowledge and practices related to traditional Mexican cuisine. As urbanization advances, there is an increased reliance on Western food consumption patterns, marked by a high consumption of sugar, salt, saturated fats, and red meat. While higher-income populations often consume animal-based foods (primarily meat), lower-income populations turn to dairy, eggs, and fish. Thus, income inequalities and resource access shape dietary preferences and purchasing power, potentially limiting the adoption of plant-based diets among lower-income populations in Monterrey. Furthermore, environmental and climate-related conditions impede the local production of food, impeding the adoption of plant-based diets. Adapted interventions addressing these multifaceted barriers, which include cultural sensitivity, and affordable plant-based alternatives are urgently needed to support the transition toward a healthier food system in this area.

## 4. Discussion

This study presents a comprehensive analysis of food access and availability distribution in Monterrey, Mexico. The findings reveal notable concentrations of fresh food outlets in the central urban zones of Monterrey, San Nicolás de los Garza, General Escobedo, and San Pedro Garza García, suggesting greater access to nutritious food. However, this accessibility declines significantly as one moves away from the urban center, notably in peripheral municipalities characterized by higher levels of poverty and marginalization, including García, El Carmen, General Zuazua, Santa Catarina, and Santiago. The study highlights the vulnerability of these marginalized communities on the metropolitan periphery, amplifying food insecurity issues. The study also identifies significant disparities in food environments across different Monterrey municipalities, including areas classified as “food deserts” with limited access to fresh and nutritious foods, and regions that demonstrate characteristics of “food swamps” where processed food outlets outnumber fresh food options.

The municipalities of García and General Zuazua pose significant challenges, as a substantial population resides there with limited food access. The findings emphasize the urgent need for targeted interventions to address food insecurity and enhance dietary quality, particularly in vulnerable communities. Furthermore, the study identifies “food oasis” municipalities where efforts to promote access to fresh and nutritious foods have been relatively successful. However, even in these areas, the prevalence of processed food outlets alongside fresh food options presents challenges. The literature indicates that Monterrey does not have a significant informal food environment compared to other major Mexican cities, further reducing the access to food for marginalized groups [37]. Moving forward, future research endeavors should prioritize the development of tailored interventions that address the specific needs and challenges faced by communities in different areas of Monterrey. By focusing on community-specific approaches, we can work toward food security and foster healthier eating habits in Monterrey.

Our analysis on diets highlights the compatibility of the traditional Mexican diet with contemporary dietary guidelines, particularly those advocated by the EAT-Lancet guidelines. This alignment underscores the potential for regional cuisine to serve as a foundation for healthier and more sustainable eating habits. Previous research [49] examined the health benefits of traditional Mexican food intake, revealing its potential to enhance overall well-being. Studies link traditional dishes to lower chronic disease risks like obesity, diabetes, and cardiovascular issues [50,51]. Additionally, research delves into the nutritional value of staple ingredients and preparations [52], consistently showing better health outcomes for those adhering to traditional diets than individuals consuming Westernized diets high in processed foods and saturated fats [29,53].

In the Noreste of Mexico, historical and socioeconomic factors fuel a strong preference for meat consumption, while natural and social barriers hinder crop production. Evidence based on dietary analysis suggests that despite the prominence of beef in traditional recipes in Nuevo León, dishes incorporating cacti as a main ingredient or salsas as part of the meat dish provide nutritional benefits, encouraging a healthier diet. Considering the nutrimental value of traditional recipes, interventions can be customized to encourage a shift toward healthier dietary patterns. Previous research has indicated that consumer behavior changes positively with a better understanding of nutrients, facilitating more informed decisions. Integrating traditional food knowledge into public health initiatives and dietary guidelines could substantially enhance population health outcomes and sustainability efforts [53]. However, further research is essential to explore effective strategies for integrating traditional ingredients and preparation into contemporary guidelines and fostering their acceptance, especially in regions where the use of plant-based diets is less prevalent.

The advance of urbanization further entrenches adherence to Western dietary practices, hindering the adoption of traditional plant-based diets. This trend is compounded by unequal food distribution and access, where unhealthy food consumption is often the result of disparities in availability and accessibility rather than insufficient production. It is imperative to address socioeconomic inequalities and cultural dietary preferences, thereby enhancing overall nutritional quality and promoting better health outcomes. By understanding food environments and their relationship with current dietary patterns along with the importance of food culture, we can better understand how tailored interventions can promote healthier eating habits in the Noreste of Mexico and preserve the local culinary heritage.

In our paper, we acknowledge several limitations. Our analysis of non-spatial barriers was limited to the literature on dietary guidelines, which may not capture the full spectrum of eating habits and practices. Research in the future could benefit from incorporating data from government databases and combining it with fieldwork to provide a more complete understanding. Additionally, regarding the dietary patterns in Nuevo León based on collected recipes, we concentrated on ingredients rather than methods of preparation or portion sizes. Future research should address these aspects and contrast them with popular dietary practices to develop culturally sensitive approaches. Future research should also explore the role of local agency in shaping eating behaviors and promoting urban agriculture, particularly within the context of cultural norms and societal influences. Finally, the spatial analysis that was conducted at the municipal scale uses aggregated data, thus it does not allow for smaller scale neighborhood-level understanding of patterns in food environments. Further research should be aimed at conducting analysis with disaggregated data at the neighborhood, or even block, scale. This small grain analysis should be focused on the critical areas identified in this research to shed light on the variety of pre-existing and developing conditions in the urban environment that hinder the availability and accessibility of fresh food and support more targeted local policies.

## 5. Conclusions

The study results revealed significant disparities in food accessibility across the entire metropolitan area of Monterrey, emphasizing the pressing need for targeted interventions to alleviate food insecurity and encourage healthier dietary practices, especially among marginalized communities. While urban and periurban agriculture offers potential solutions to food desert challenges, tailored strategies are crucial in areas constrained by soil and rainfall limitations. Addressing food insecurity through urban agriculture requires careful consideration of each municipality’s unique conditions and challenges. Moreover, transitioning to a plant-based diet in Monterrey presents multifaceted challenges influenced by socioeconomic status, biocultural diversity, dietary trends, and urbanization. These factors highlight the importance of addressing not only physical barriers to accessing fresh and nutritious foods but also socio-cultural and economic factors shaping dietary behaviors and preferences.

The traditional Mexican diet provides a robust foundation for adhering to dietary guidelines like the EAT diet in Mexico. However, obstacles such as limited urban space for crop cultivation and ingredient accessibility must be addressed. This underscores the urgency of tackling unequal food distribution and access, significant contributors to unhealthy dietary patterns amid the urbanization and globalization processes. Further research into these interconnected factors is imperative for devising effective strategies to implement healthy dietary guidelines aligned with environmental limitations. Policymakers can address food insecurity in urban areas by understanding complex food access and distribution issues. Tailored interventions that tackle the root causes of this problem and promote sustainable and culturally appropriate eating practices are urgently needed. In future policy, it is essential to take a multidimensional approach that considers traditional dietary patterns, contemporary nutritional guidelines, and the challenges posed by the cultural and environmental characteristics of Noreste Mexico.

## Figures and Tables

**Figure 1 nutrients-16-01259-f001:**
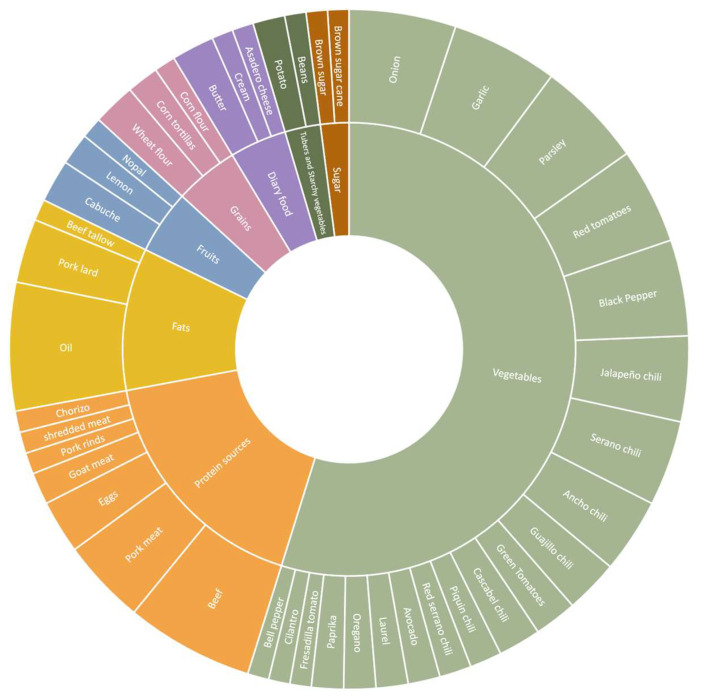
Collected recipes from interviews with female cooks and chefs, by authors.

**Figure 2 nutrients-16-01259-f002:**
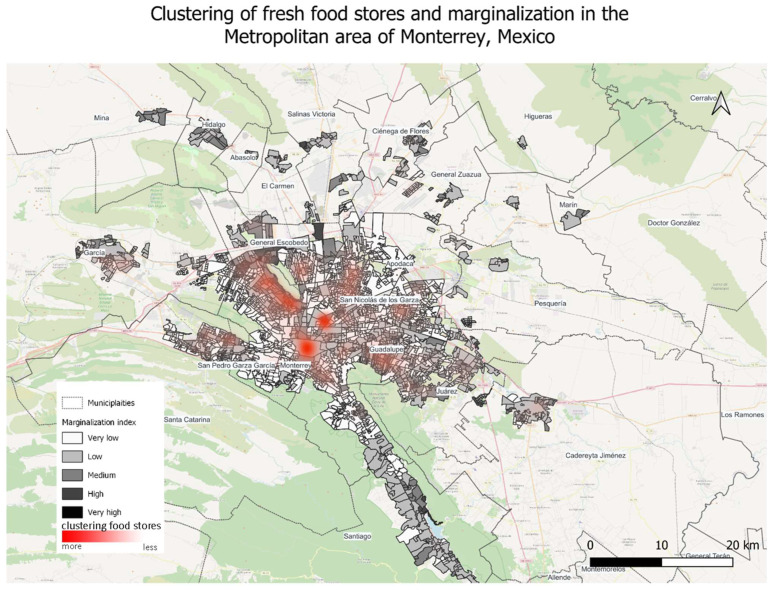
Fresh food clusters in the metropolitan area of Monterrey, by authors.

**Figure 3 nutrients-16-01259-f003:**
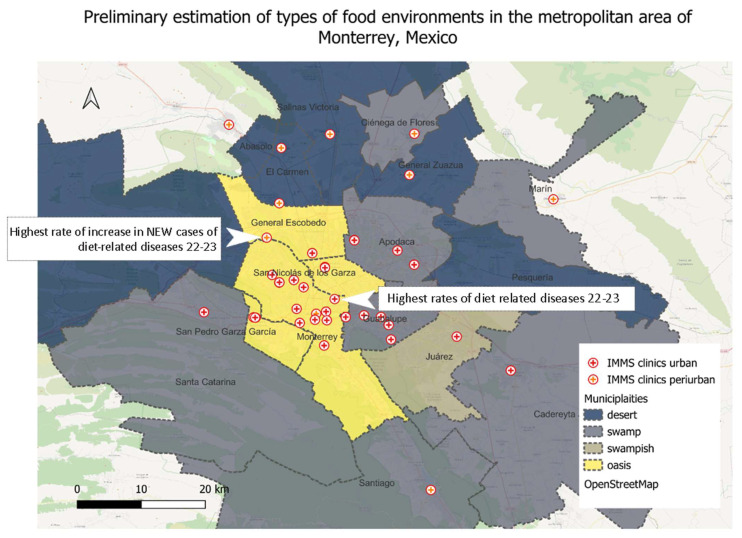
The prevalence of diet-related diseases in different food environments in Monterrey.

**Figure 4 nutrients-16-01259-f004:**
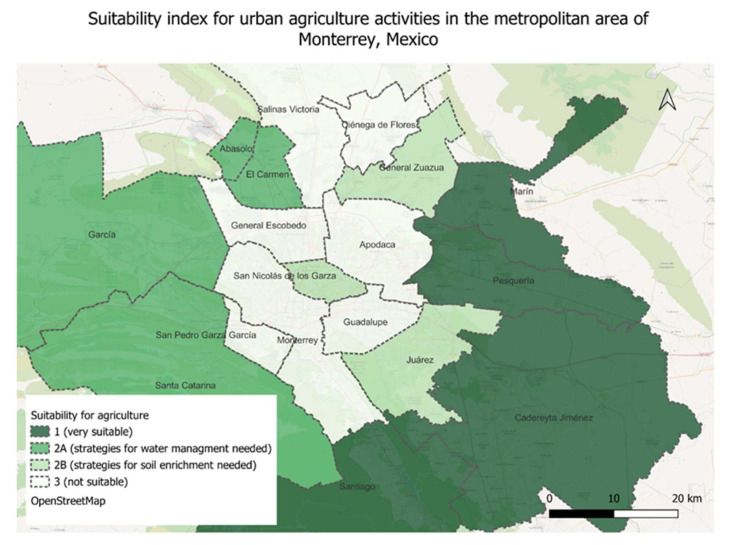
Suitability of urban and periurban areas in Monterrey for plant-based local food production.

**Figure 5 nutrients-16-01259-f005:**
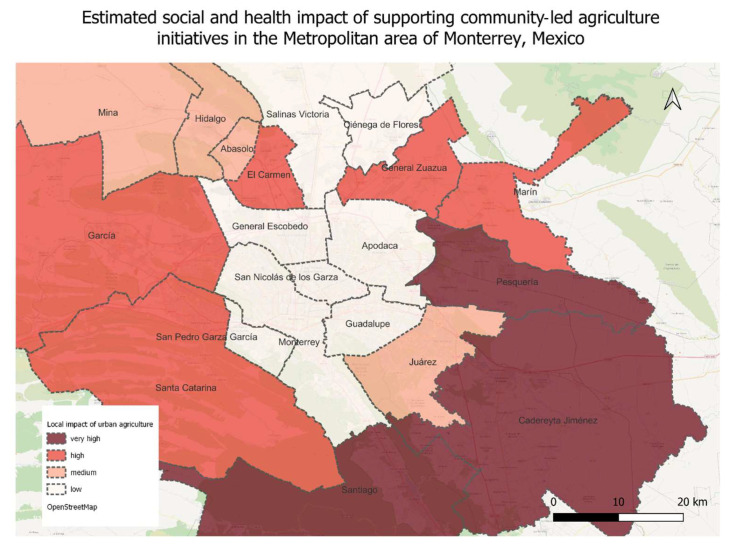
Mapping the impact of local food production in the metropolitan area of Monterrey.

**Figure 6 nutrients-16-01259-f006:**
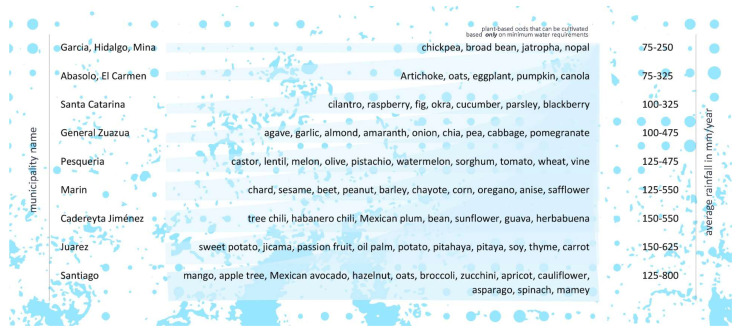
Plant-based foods that could be promoted for urban agriculture in each municipality, based on [46].

**Table 1 nutrients-16-01259-t001:** The traditional Mexican diet, by authors.

Category	Foodstuffs	Consumption Level
Whole Grains	Maize, wheat (such as bread and pasta), amaranth	High
Tubers and Starchy Vegetables	Potatoes, sweet potatoes, yucca	High
Vegetables	Squash, tomatoes, onions, chayote, nopales, maguey, jicama	High
Fruits	Capulin, tejocote, pineapple, citrus fruits (e.g., oranges, lemons, limes), papaya	Moderate (Central Mexico) to High (Southern Mexico)
Protein Sources	Turkey, chicken, ducks, venison, beef, insects (e.g., grasshoppers and locusts)	Low
Fats	Avocado, peanuts	Low
Sugars	Honey, sugar, sugarcane	Low
Dairy Food	Not typically included	None

By authors based on https://www.ncbi.nlm.nih.gov/pmc/articles/PMC10011591/, accessed on 21 February 2024.

**Table 2 nutrients-16-01259-t002:** EAT-Lancet intake recommendations for a healthy diet.

Category	Subcategory	Suggested Portion Size of Foods Most Consumed in Mexico
Grains	Whole Grains	1 corn tortilla or its equivalent in corn dough or flour
	Other high-fiber grains	4 tablespoons of raw oats or amaranth, 10 tablespoons of wheat bran, 1/2 slice of whole bread, 1/2 cup of whole pasta or brown rice
	Refined grains	1/2 cup of cooked pasta or rice, 1/2 slice of white bread, 1/2 bolillo
	Grains with excess added sugar or saturated fat	1/3 of sweet bread, 2–3 sweet cookies, 1/3 of a small chips bag, 1 flour tortilla
Tubers and Starchy Vegetables	Potatoes	1/2 potato
Fruits	All fruits	1/2–1 cup of banana, papaya, orange, apple, pear, pineapple, guava
Vegetables	All vegetables	1/2–1 cup of tomato, onion, carrot, green, tomato, zucchini, nopales, chilies, squash
Dairy Food	Whole milk or derivative equivalents (e.g., cheese)	2/3 cup of chocolate milk, milk with coffee and sugar, or fruit smoothie with milk; ½ cup of sweetened yogurt
Protein Sources	Red meat	60 g red meat (beef and pork)
	Processed meat	1 sausage or 1 slice of ham, 16 g (1 tablespoon) of chorizo
	Chicken and other poultry	120 g or 1 chicken leg, thigh, or breast
	Eggs	1 Egg
	Fish	120 g of fish, 20 shrimps, 1 can of tuna
	Legumes	1/2 cup of cooked beans, lentils, chickpeas, or broad beans
	Nuts	20 g or 4 tablespoons of peanuts, walnuts, almonds, or seeds (chia, sesame, pumpkin, or sunflower)
Added fats	Plant oils	1 tablespoon of vegetable oil (corn, sunflower, soy, or canola) or mayonnaise
	Lard or tallow	1 tablespoon of lard
Added sugars	All sweeteners	1 cup of sugar-sweetened beverage, 2 tablespoons of sugar, honey, condensed milk, or catsup, 1/2 cup of jello, 15 g of chocolate, 1/3 cup of ice cream

Based on https://www.sciencedirect.com/science/article/pii/S0002916522006931, accessed on 22 February 2024.

**Table 3 nutrients-16-01259-t003:** Estimation of the types of food environments based on availability and accessibility of fresh food.

	Municipality	Population *	No. Food Stores	No. Fresh Food Stores	No. Processed Food Stores	Pop/Store	Walkability to Store **	Type of Food Environment
urban	Apodaca	656,464	5285	2654	2631	98	30–40 min	swamp
urban	Cadereyta Jiménez	122,337	1252	670	582	98	20–30 min	swamp
urban	El Carmen	104,478	236	148	88	443	30–40 min	desert
urban	Ciénega de Flores	68,747	526	312	214	131	30–40 min	swamp
urban	García	397,205	2425	1538	887	164	20–30 min	desert
urban	San Pedro Garza García	132,169	1437	790	647	92	10–20 min	oasis
urban	General Escobedo	481,213	4533	2531	2002	106	10–20 min	oasis
urban	General Zuazua	102,149	620	405	215	165	50–60 min	desert
urban	Guadalupe	643,143	7421	3576	3845	87	15–25 min	swamp
urban	Juárez	471,523	3666	2167	1499	129	20–30 min	swampish
urban	Monterrey	1,069,238	15,275	7986	7289	70	7–17 min	oasis
urban	Pesquería	147,624	512	356	156	288	35–45 min	desert
urban	Salinas Victoria	86,766	403	269	134	215	40–50 min	desert
urban	San Nicolás de los Garza	412,199	5180	2657	2523	80	10–20 min	oasis
urban	Santa Catarina	306,322	2810	1540	1270	109	35–45 min	swamp
urban	Santiago	46,784	598	327	271	78	25–35 min	swamp
Total		5,322,117	52179	27,926	24,253			
periurban	Hidalgo	16,086	239	123	116	67	35–45 min	swamp
periurban	Abasolo	2974	23	17	6	129	50–60 min	desert
periurban	Marin	4719	54	29	25	87	no data	swamp
periurban	Mina	6048	78	50	28	78	no data	ND
Total		29,827	394	219	175			

* based on the census data INEGI 2020, https://www.inegi.org.mx/programas/ccpv/2020/, accessed on. ** based on the estimate from Observatorio de Ciudades, Tec de Monterrey, available at: https://observatorio-ciudades.github.io/visor-15minutos/cities/monterrey.html, accessed on 23 February 2024.

**Table 4 nutrients-16-01259-t004:** Estimation of possible cultivation areas in ZMM.

	Municipality	Total Area km^2^ *	No. Houses *	Unbuilt Area km^2^ **	Natural Areas/Slopes	Possible Cultivation Area km^2^ **	Dominant Soil Type ***	Volume Range Dry/Wet Season in mm ***	Runn-Off Index ***	Suitability for Urban Agriculture ****
urban	Apodaca	224.0	208,426	115.1	20%	92.1	lixisol	100–475	5%	3
urban	Cadereyta Jiménez	1140.9	44,111	789.8	5%	750.3	andosol	150–550	10%	1
urban	El Carmen	104.3	38,067	65.4	10%	58.9	phaozem/leptisol	75–325	5%	2A
urban	Ciénega de Flores	138.7	30,004	91.1	0%	91.1	calcisol	100–475	10%	3
urban	García	1032.0	132,710	695.9	50%	347.9	chernozem	75–250	5%	2A
urban	San Pedro Garza García	70.8	41,193	41.3	20%	40.1	gipsisol	100–475	10%	3
urban	General Escobedo	149.4	152,111	74.2	20%	59.3	gipsisol	100–400	20%	3
urban	General Zuazua	184.5	39,825	121.2	3%	117.5	kastanozem	100–475	5%	2B
urban	Guadalupe	118.4	207,099	41.5	20%	33.2	durisol	125–550	10%	3
urban	Juárez	247.3	175,293	138.1	30%	96.6	leptisol	150–625	20%	2B
urban	Monterrey	324.4	368,780	153.3	40%	92.0	ND-urban	100–475	10%	3
urban	Pesquería	322.8	51,612	215.6	10%	194.1	andosol	125–475	10%	1
urban	Salinas Victoria	1667.4	3,2149	1160.8	40%	696.5	calcisol	75–400	5%	3
urban	San Nicolás de los Garza	60.2	133,725	15.4	5%	14.6	acrisol	100–475	10%	2B
urban	Santa Catarina	915.8	89,586	623.1	90%	62.3	phaozem/cambisol	100–325	5%	2A
urban	Santiago	739.2	17,602	513.9	90%	51.4	cambisol	125–800	10%	1
Total		7440.4	176,2293	4855.6		2797.9	total km^2^ of prime area 1			995.8
periurban	Hidalgo	208.0	5879	144.4	90%	14.4	phaozem/leptisol	75–250	5%	2A
periurban	Abasolo	47.5	795	33.1	20%	26.4	phaozem/leptisol	75–325	5%	2A
periurban	Marin	129.0	1339	90.0	20%	72.0	cambisol/andosol	125–550	10%	1
periurban	Mina	3915.8	1818	2740.7	10%	2466.6	cambisol/andosol	75–250	5%	2A
Total		4300.3	9831	3008.2		2579.5	total km^2^ of prime area 1			72.0

* based on the census data INEGI 2020, https://www.inegi.org.mx/programas/ccpv/2020/, accessed on 29 February 2024. ** calculated as: the total area − 30% of total area (roads & communication) − total houses × 100 m^2^ (medium housing area) × 2 (all other land uses, based on van der Hoek’s (2008, [39]) mixed-use index, which suggests that the highest level of mix is achieved when residential floor space in a given territory equals that of all other functions. *** SIATL tool, INEGI: https://antares.inegi.org.mx/analisis/red_hidro/siatl/#, accessed on 29 February 2024. **** 1—very high; 2A—high, water scarcity; 2B—high, poor soil quality; 3—low, water scarcity and poor soil quality.

**Table 5 nutrients-16-01259-t005:** Estimation of impact of urban/periurban agriculture in Monterrey.

	Municipality	Population *	Indigenous Pop. % *	% Pop with Social Security *	Moderate and Extreme Poverty % *	Type of Food Environment	Social and Health Impacts **
urban	Apodaca	656,464	0.93	55.30	14.20	swamp	Low
urban	Cadereyta Jiménez	122,337	1.98	47.70	21.12	swamp	Very high
urban	El Carmen	104,478	3.57	65.30	28.49	desert	High
urban	Ciénega de Flores	68,747	2.87	62.00	23.68	swamp	Low
urban	García	397,205	3.47	59.20	22.62	desert	High
urban	San Pedro Garza García	132,169	1.65	24.80	5.45	oasis	Low
urban	General Escobedo	481,213	1.79	49.90	24.96	oasis	Low
urban	General Zuazua	102,149	3.45	66.00	23.85	desert	High
urban	Guadalupe	643,143	0.66	54.00	15.78	swamp	Low
urban	Juárez	471,523	2.14	60.10	24.45	swampish	Medium
urban	Monterrey	1,069,238	1.29	49.40	19.27	oasis	Low
urban	Pesquería	147,624	6.54	64.30	23.47	desert	Veryhigh
urban	Salinas Victoria	86,766	3.66	60.20	27.34	desert	Low
urban	San Nicolás de los Garza	412,199	0.36	57.00	10.87	oasis	Low
urban	Santa Catarina	306,322	1.51	60.80	16.43	swamp	High
urban	Santiago	46,784	1.59	55.20	12.43	swamp	Very high
Total		5,322,117	37.46				
periurban	Hidalgo	16,086	0.32	66.70	23.25	swamp	Medium
periurban	Abasolo	2974	0.40	59.70	10.37	desert	Medium
periurban	Marin	4719	ND	ND	ND	swamp	High
periurban	Mina	6048	ND	ND	ND	ND	Medium
Total		29,827	0.72				

* based on the census data INEGI 2020, https://www.inegi.org.mx/programas/ccpv/2020/, accessed on 29 February 2024. ** estimated based on the suitability for agriculture index in table, population served, area available for farming, and poverty rate.

## Data Availability

The original contributions presented in the study are included in the article, further inquiries can be directed to the corresponding author.

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
