# Peer review of "Spatial Barriers to Transforming toward a Healthy Food System in the Noreste of Mexico"

_nutrients, 2024, doi:10.3390/nu16091259_

Round 1
Reviewer 1 Report
Comments and Suggestions for Authors
Dear Authors,
A well written and important contribution to the field.
There are a couple of references I would like you to consider including which relate to the work of Clapp et al and the High Level Panel of Expert (HLPE) in Food and Nutrition and the 6 dimensions of food in/security including availability, access, utilisation, stability, sustainability and agency. I note that you refer to food access and availability however I was wondering if you had considered the addition of sustainability and agency within your food in/security discussion. I'm particularly interested in the community agency perspective and the voice of marginalised community members having their voice heard through authentic participation in decision-making processes which could be a recommendation. I will leave it to you to decide if it fits in.
Ref 1 - Clapp et al 2022, Viewpoint: The case for a six-dimensional food security framework - ScienceDirect
Ref 2 - HLPE 2020, Food Security and Nutrition: Building a global narrative towards 2030 1_DeRHbiFK7ysUizHuY4rsFw.png (700×649) (scalingupnutrition.org)
I have some suggestions which you might consider for improving the clarity of your results: 1), Integration of findings with literature review - perhaps a more seamless connection between the results and the literature in the introduction to help the reader to understand how the new findings build on existing knowledge or diverge, 2) Highlight the key findings - whilst the results are detailed, perhaps provided a summary of key findings, at the beginning or end of each section.
Overall the discussion and conclusion are written well. A couple of suggestions you could consider 1) provide more detailed recommendations for policy and practice based on the study's findings though this may be out of scope 2) discuss the limitation of the study more explicitly to provide context for the findings and suggestions for future research.
regards
Reviewer
Author Response
Dear reviewer,
Thank you for your valuable and positive review of our article. Below we explain point by point how we have revised the article based on your comments.
A well written and important contribution to the field. Thank you.
There are a couple of references I would like you to consider including which relate to the work of Clapp et al and the High Level Panel of Expert (HLPE) in Food and Nutrition and the 6 dimensions of food in/security including availability, access, utilisation, stability, sustainability and agency. I note that you refer to food access and availability however I was wondering if you had considered the addition of sustainability and agency within your food in/security discussion. I'm particularly interested in the community agency perspective and the voice of marginalised community members having their voice heard through authentic participation in decision-making processes which could be a recommendation. I will leave it to you to decide if it fits in.
Ref 1 - Clapp et al 2022, Viewpoint: The case for a six-dimensional food security framework - ScienceDirect
Ref 2 - HLPE 2020, Food Security and Nutrition: Building a global narrative towards 2030 1_DeRHbiFK7ysUizHuY4rsFw.png (700×649) (scalingupnutrition.org)
Thank you for your thoughtful suggestions and the reference material you provided. It has been very insightful. We included in the paper a paragraph addressing sustainability and agency.
I have some suggestions which you might consider for improving the clarity of your results: 1), Integration of findings with literature review - perhaps a more seamless connection between the results and the literature in the introduction to help the reader to understand how the new findings build on existing knowledge or diverge, 2) Highlight the key findings - whilst the results are detailed, perhaps provided a summary of key findings, at the beginning or end of each section.
Overall the discussion and conclusion are written well. A couple of suggestions you could consider 1) provide more detailed recommendations for policy and practice based on the study's findings though this may be out of scope 2) discuss the limitation of the study more explicitly to provide context for the findings and suggestions for future research.
Thank you for your feedback. In response, we have revised the discussion section to include a more explicit discussion of the limitations of the study.
Reviewer 2 Report
Comments and Suggestions for Authors
The manuscript reports findings from a very ambitious and multicomponent study that explores spatial and non-spatial characteristics of an area in Mexico that may impede adoption of healthful diets. As such, it is timely and of interest to the nutrition, food systems and food environment research communities. Most of the manuscript is very well written, except for the methods which are sometimes difficult to follow due to switching between past and present verb tenses. Also, the results often contain discussion points, although I am not sure how the authors can avoid that as their discussion is quite extensive in its present form. The tables and figures are well down with the minor exceptions noted in the specific comments that follow. My compliments to the authors on a well written, albeit massive, paper that comprehensively explores spatial and nonspatial characteristics of the food landscape in Noreste Mexico. Specific minor suggestions to improve the clarity and readability of the manuscript follow.
Abstract
1. Line 20: Suggest replacing “aimed at” with “focusing on” to avoid using “aims” and “aimed at” in same sentence.
Introduction
1. Line 71: Suggest authors replace “non-communicable diseases” with acronym “NCDs” that was previously defined (line 41).
2. Line 88: Suggest authors are consistent with spelling of “sociocultural” vs “socio-cultural” throughout manuscript.
3. Line 93: Suggest deleting “a” before “long-term.”
4. Lines 98-99: For clarity, suggest revising as “Only a minimal proportion of Mexican adults regularly consumes…follows less healthy Western…”
5. Line 101: Suggest replacing “negative unhealthy outcomes” with “chronic conditions.”
6. Line 109: Replace “consists in” with “consists of” for clarity.
Materials & Methods
7. General comment: Section should be written in past tense; currently, mix of past and present tense.
8. Line 133: Suggest deleting “hypothetical” as these 3 diets/dietary patterns actually exist.
9. Line 144: Sentence does not make sense.
10. Line 160: Suggest authors define or give examples of what they mean with the term “informal markets.”
11. Lines 202-209: The 3 classifications do not seem to cover all scenarios. For example, how are areas with unbuilt land less than the built area with good soil for cultivation and sufficient water supply classified?
Results
12. Line 231: “Several” should not be capitalized.
13. Line 254: The acronym EAT-HRD requires definition.
14. Line 270: The acronym MDGs requires definition.
15. Lines 274-275: For noun-verb match, replace “require” with “requires.”
16. Lines 357: What type of insecurity?
17. Line 362: Per-urban is misspelled (should be peri-urban). Also, authors need to be consistent throughout manuscript in format for this word (either peri-urban or periurban).
18. Line 364: What do the authors mean by “walkable access is extremely difficult”?
19. Line 365: Suggest authors are specific in their terminology and preface “swamp” with “food.” Also, the definition presented here for food swamp is not the same as the one the authors presented in the methods.
20. Lines 367-368: Authors need to provide definition of “extreme swamp conditions” in the methods.
21. Lines 371-373: Authors need to provide definition of “swampish” in the methods. Also, the municipality cannot have both low and medium accessibility.
Discussion
22. Lines 556-558: Sentence requires a reference.
23. Line 567: Is it availability or accessibility as shown in the current study?
24. Line 570: Replace “its” with “their” (environments is plural) and “between” with “with.”
25. Line 594: If by “they” authors are referring to policymakers, sentence needs to be revised as policymakers set policy (not conduct interventions).
26. Line 598-599: Suggest deleting this sentence as it is superfluous. Preceding sentence provides much stronger concluding sentence.
Figure 1
27. While this figure appears quite colorful and potentially useful, the font is too small for readability and the color scheme is not defined.
Figure 2
28. This figure appears very well done and potentially informative, but the font size is too small for readability.
Table 1
29. Label for third column is misspelled (should be Consumption).
30. Last row, foodstuffs, “in the traditional Mexican diet” should be deleted (redundant with table title).
31. There is a “star” footnote below the table with no corresponding “star” in the table.
Table 2
32. For Fruits row, examples in third column appear to be mix of fruits and vegetables.
33. For Vegetables row, examples in third column all appear to be fruit.
Comments on the Quality of English Language
No issues with the exception of mixed (past and present) verb tense in the methods.
Author Response
Dear reviewer,
We thank you very much for your constructive and positive review. Below you will find our responses to your comments, point by point.
The manuscript reports findings from a very ambitious and multicomponent study that explores spatial and non-spatial characteristics of an area in Mexico that may impede adoption of healthful diets. As such, it is timely and of interest to the nutrition, food systems and food environment research communities. Most of the manuscript is very well written, except for the methods which are sometimes difficult to follow due to switching between past and present verb tenses.
Answer: Thank you, we have adjusted the past/present tense where possible.
Also, the results often contain discussion points, although I am not sure how the authors can avoid that as their discussion is quite extensive in its present form. The tables and figures are well down with the minor exceptions noted in the specific comments that follow. My compliments to the authors on a well written, albeit massive, paper that comprehensively explores spatial and nonspatial characteristics of the food landscape in Noreste Mexico. Specific minor suggestions to improve the clarity and readability of the manuscript follow.
Abstract
- Line 20: Suggest replacing “aimed at” with “focusing on” to avoid using “aims” and “aimed at” in same sentence.
Introduction
- Line 71: Suggest authors replace “non-communicable diseases” with acronym “NCDs” that was previously defined (line 41).
- Line 88: Suggest authors are consistent with spelling of “sociocultural” vs “socio-cultural” throughout manuscript.
- Line 93: Suggest deleting “a” before “long-term.”
- Lines 98-99: For clarity, suggest revising as “Only a minimal proportion of Mexican adults regularly consumes…follows less healthy Western…”
- Line 101: Suggest replacing “negative unhealthy outcomes” with “chronic conditions.”
- Line 109: Replace “consists in” with “consists of” for clarity.
Materials & Methods
- General comment: Section should be written in past tense; currently, mix of past and present tense.
- Line 133: Suggest deleting “hypothetical” as these 3 diets/dietary patterns actually exist.
- Line 144: Sentence does not make sense.
- Line 160: Suggest authors define or give examples of what they mean with the term “informal markets.”
- Lines 202-209: The 3 classifications do not seem to cover all scenarios. For example, how are areas with unbuilt land less than the built area with good soil for cultivation and sufficient water supply classified?
Results
- Line 231: “Several” should not be capitalized.
- Line 254: The acronym EAT-HRD requires definition.
- Line 270: The acronym MDGs requires definition.
- Lines 274-275: For noun-verb match, replace “require” with “requires.”
- Lines 357: What type of insecurity?
- Line 362: Per-urban is misspelled (should be peri-urban). Also, authors need to be consistent throughout manuscript in format for this word (either peri-urban or periurban).
- Line 364: What do the authors mean by “walkable access is extremely difficult”?
- Line 365: Suggest authors are specific in their terminology and preface “swamp” with “food.” Also, the definition presented here for food swamp is not the same as the one the authors presented in the methods.
- Lines 367-368: Authors need to provide definition of “extreme swamp conditions” in the methods.
- Lines 371-373: Authors need to provide definition of “swampish” in the methods. Also, the municipality cannot have both low and medium accessibility.
Discussion
- Lines 556-558: Sentence requires a reference.
- Line 567: Is it availability or accessibility as shown in the current study?
- Line 570: Replace “its” with “their” (environments is plural) and “between” with “with.”
- Line 594: If by “they” authors are referring to policymakers, sentence needs to be revised as policymakers set policy (not conduct interventions).
- Line 598-599: Suggest deleting this sentence as it is superfluous. Preceding sentence provides much stronger concluding sentence.
Figure 1
- While this figure appears quite colorful and potentially useful, the font is too small for readability and the color scheme is not defined.
Figure 2
- This figure appears very well done and potentially informative, but the font size is too small for readability.
Table 1
- Label for third column is misspelled (should be Consumption).
- Last row, foodstuffs, “in the traditional Mexican diet” should be deleted (redundant with table title).
- There is a “star” footnote below the table with no corresponding “star” in the table.
Table 2
- For Fruits row, examples in third column appear to be mix of fruits and vegetables.
- For Vegetables row, examples in third column all appear to be fruit.
Answer: Thank you, for the thorough revisions, we have accepted and corrected all of your suggestions. You can check the revised version attached.
Reviewer 3 Report
Comments and Suggestions for Authors
This interesting paper classifies the study areas into food deserts, swamps, and oases. It examines the socioeconomic conditions and potential for promoting urban and peri-urban agriculture in each area. The reviewer believes the paper would have been more substantial if the authors had surveyed residents regarding either 1) or 2) below and presented the analysis results using the data. While the paper makes good use of the points made in previous studies in its discussion, it is unsurprising that some readers may feel that it merely organizes census data. The explanations for each need to be more profound. However, despite these shortcomings, the paper is worthy of publication because it is a research result that will be helpful to those in charge of administration in each country, although the following 3) need to be addressed.
1) It needs to be clarified how the collected recipes can be aggregated to draw the pie chart in Figure 1. Even if the recipes are collected, it is only possible to know the nutritional status of the residents if we know what kind and how much food they eat.
2) Equally important, the paper lacks detailed information on the current state of urban agriculture, such as the extent of its practice, the types of crops being grown, and the percentage of households involved. This information is crucial for formulating effective policies and strategies for urban and peri-urban agriculture.
3) The pie chart in Figure 1 is presented in a small font, making it challenging to read. A larger font size would significantly improve its readability, aiding in a better understanding and interpretation of the data.
Author Response
Dear reviewer,
Thank you so much for your valuable comments. We have used all point to improve the paper further. Below you can find our responses to your comments.
This interesting paper classifies the study areas into food deserts, swamps, and oases. It examines the socioeconomic conditions and potential for promoting urban and peri-urban agriculture in each area. The reviewer believes the paper would have been more substantial if the authors had surveyed residents regarding either 1) or 2) below and presented the analysis results using the data. While the paper makes good use of the points made in previous studies in its discussion, it is unsurprising that some readers may feel that it merely organizes census data. The explanations for each need to be more profound. However, despite these shortcomings, the paper is worthy of publication because it is a research result that will be helpful to those in charge of administration in each country, although the following 3) need to be addressed.
1) It needs to be clarified how the collected recipes can be aggregated to draw the pie chart in Figure 1. Even if the recipes are collected, it is only possible to know the nutritional status of the residents if we know what kind and how much food they eat.
- A more detailed explanation regarding the collected recipes has been added to our methodology section to address your concern.
2) Equally important, the paper lacks detailed information on the current state of urban agriculture, such as the extent of its practice, the types of crops being grown, and the percentage of households involved. This information is crucial for formulating effective policies and strategies for urban and peri-urban agriculture.
- In response to your feedback, we have added a paragraph regarding the state of urban agriculture today.
3) The pie chart in Figure 1 is presented in a small font, making it challenging to read. A larger font size would significantly improve its readability, aiding in a better understanding and interpretation of the data.
- Thank you for highlighting the font size issue in Figure 1. I've taken your feedback into account and have made the necessary adjustments to improve the readability of the figure.